# Prevalence and associations of anisometropia with spherical ametropia, cylindrical power, age, and sex, based on 134,603 refractive surgery candidates

**Mona Deuchert** [1], **Andreas Frings** [1]*, **Vasyl Druchkiv**[2,3], **Jakob Schweighofer**[4], **Sajjad Muhammad**[5], **Stephan Linke**[6], **Toam Katz**[2,7]

1 Department of Ophthalmology, Medical Faculty and University Hospital Duesseldorf, Heinrich Heine University, Duesseldorf, Germany, 2 Department of Ophthalmology, University Medical Center Hamburg-Eppendorf, Hamburg, Germany, 3 Research & Development Department, Clinica Baviera Valencia, Valencia, Spain, 4 Department of Ophthalmology, Vienna General Hospital, Vienna, Austria, 5 Department of Neurosurgery, Medical Faculty and University Hospital Duesseldorf, Heinrich Heine University, Duesseldorf, Germany, 6 Zentrumsehstaerke, Hamburg, Germany, 7 CARE Vision, Hamburg, Germany

* andreas.frings@med.uni-duesseldorf.de

**Data Availability Statement:** All relevant data are within the manuscript and its Supporting Information files.

## Abstract

### Purpose

To examine the prevalence and associations of anisometropia with spherical ametropia, cylindrical power, age, and sex.

### Methods

Anisometropia was analyzed for subjective refraction. In total, 134,603 refractive surgery candidates were included in the period from 2010 to 2020 at the CARE Vision Refractive Centers in Germany. Our study was approved by the local ethics committee at the University in Duesseldorf (approval date: February 9, 2021) and conducted according to the tenets of the Declaration of Helsinki and Good Clinical Practices Guidelines. The treatment contract included explicit patient consent to use medical data for scientific purposes. Correlations between anisometropia and explanatory variables were analyzed using the chi-squared test ($\chi^2$ test), nonparametric Kruskal–Wallis or Mann–Whitney U-tests, and binomial logistic regression. Power vector analysis was applied for further analysis of cylindrical power.

### Results

The median level of anisometropia ($A_{subj}$) in the whole population was 0.38 D. The prevalence of $A_{subj}$ was 17.9%. In hyperopes, all explanatory variables (spherical ametropia, cylindrical power, age, sex) were independently associated with anisometropia. $A_{subj}$ decreased with increasing cylindrical ametropia: an increase in cylindrical power by 7.8 D reduced the chance of observing anisometropia by half. It was also associated with male sex. $A_{subj}$ decreased by half with a 16.7 D increase in spherical power and an increase in age by 22.7 years. In myopes, cylindrical power was most strongly associated with anisometropia: an

**Funding:** The author(s) received no specific funding for this work.

**Competing interests:** The authors have declared that no competing interests exist.

increase in (negative) cylindrical power by 2.15 D doubled the chances of observing anisometropia in myopes. In addition, advancing age (double chance with an increase of 38.3 years), increasing spherical power (double chance with an increase of 8.15 D), and female sex correlated positively with increasing anisometropia in myopes.

## Conclusions

This retrospective analysis gives evidence for the independent association between anisometropia and spherical power, cylindrical power, age, and sex in myopic refractive surgery candidates. The relation of anisometropia with age was positive in myopes but negative in hyperopes. The analysis of sex revealed a positive relation of female sex and anisometropia in myopes and furthermore revealed a positive relation of male sex and anisometropia in hyperopes. Further clinical research into the underlying mechanisms behind our findings is indicated.

## Introduction

Anisometropia is a difference between the refractive power of the left and right eyes. There are two types: physiological and higher anisometropia. The latter can cause amblyopia and strabismus [1, 2]. Given these clinically relevant pathologies, it is of great interest and clinical relevance to carry out research in the field of anisometropia.

According to Weale [3], a difference of up to 1 diopter (D) is generally not considered to be anisometropia. The correlation between right and left eye refractive error is generally high, as shown in the review carried out by Armstrong [4]—e.g., with Rosner [5] presenting an estimated correlation between the spherical refractive error of the right and left eyes of an individual of 0.943. Nunes et al. [6] presented a comprehensive overview of the literature on anisometropia and found that the prevalence ranges from 1.0% to 18.1% depending on age, the geographical location of the sample population, and the type of refractive test used (retinoscopy vs. cycloplegic autorefraction). Several studies have focused on childhood studies [7–10], adult populations as follow-up studies to childhood studies [11, 12], from geriatric patients [13], from data on patients visiting optometry practices [14, 15], and from refractive surgery candidates [16, 17].

Several studies have reported statistical associations between anisometropia and spherical ametropia, cylindrical ametropia, and age. Only three studies known to the authors have explored the independent associations among anisometropia, spherical ametropia, cylindrical ametropia, and age [7, 14, 15]. This is noteworthy given the association between spherical and cylindrical refractive errors as reported by Guggenheim and Farbrother [18]. Tong et al. [7] found that myopia, age, and sex were independently associated with anisometropia in a study of Singaporean schoolchildren, while Qin et al. [14] reported an independent link between anisometropia and both spherical and cylindrical ametropia in a sample of 90,884 optometry patients in the UK.

This study is an analysis of 134,603 refractive surgery candidates. We aim to examine the prevalence of anisometropia in this population and its independent association with spherical ametropia, cylindrical ametropia, age, and sex. A review of the literature has not yielded previous studies with larger data sets. This study builds on and extends the work carried out by

Linke et al. [16], which included data from 13,535 individuals and which was itself based on the work carried out by Qin et al. [14] with 90,884 individuals.

Because of the focus on amblyopia and strabismus, a large portion of the studies in the field of anisometropia is centered on school-aged children, whereas the investigation of adult individuals has seen less research activity. However, important clinical findings can also be derived from the consideration of the anisometropia risk factors *spherical ametropia*, *cylindrical ametropia*, *age*, *and sex* in adulthood. We thus aim to contribute statistically robust results to ongoing research on anisometropia in adults.

## Methods

Patients who visited a multicenter chain of refractive clinics in Germany between January 2010 and January 2020 for the correction of their ametropia were considered for inclusion in this study. Most of the individuals were eligible for either laser assisted in situ keratomileusis (LASIK) or photorefractive keratectomy (PRK) laser refractive surgery. Others who exceeded the range for laser correction were eligible for phakic intraocular lens surgery or clear lens extraction. Patients with previous surgical interventions, such as cataract surgery, or patients with eye conditions were not included in the study.

Fully anonymized data was extracted for retrospective analysis from the patient data base on July 16, 2022. The authors had no access to information that could identify individual participants during or after data collection. The study was approved by the local ethics committee (Heinrich-Heine University, Dusseldorf, Germany) on February 9, 2021 (#2021–1278) and adhered to the principles of the Declaration of Helsinki and the GDPR (General Data Protection Regulation). All patients voluntarily provided written informed consent to use medical data for scientific purposes during the surgery recruitment process.

Next to the determination of subjective and (for parts of the study population) cycloplegic refraction, a comprehensive examination was conducted for each patient to ensure that the preconditions for refractive surgery were met. The examination included a general ophthalmological evaluation using the slit lamp, medical and ophthalmic history, and preoperative measurements such as uncorrected distance visual acuity (UDVA), corrected distance visual acuity (CDVA), tonometry (using non-contact tonometers), pupillometry, corneal topography, and pachymetry (using either Orbscan or Pentacam). Patients unqualified for refractive surgery after the preoperative examinations were excluded from the study.

To avoid confusion during analysis, all refractive data was converted to minus cylinder form.

### Classification criteria

To best compare the findings of this research with those of the aforementioned, less extensive study carried out by Linke et al. [16], the classification criteria were adopted from the latter. The study population was divided into two groups: the myopic group, consisting of individuals with a spherical equivalent (SE) of <0 D or those with mixed astigmatism (SE < 0 D) in the less ametropic eye; and the hyperopic group, consisting of individuals with SE >0 D in the less ametropic eye. Anisometropia was defined as the absolute difference in SE power between the right and left eyes. Anisometropia was calculated separately for subjective and cycloplegic refraction, and each eye was categorized into one of four groups: non-anisometropia (<1.00 D); mild anisometropia (1.00 ≤ D ≤ 1.99 D); moderate anisometropia (2.00 ≤ D ≤ 2.99); and severe anisometropia (≥3.00 D).

The subjects were further divided into one of six age categories: group 1, <20 years old; group 2, 20–29 years old; group 3, 30–39 years old; group 4, 40–49 years old; group 5, 50–59

years old; and group 6, $\geq$60 years old. In addition, analyses were performed on only those subjects aged between 20 and 40 years.

The vectorial approach (J0, J45) was first described by Thibos et al. [19]. Subjective refractions in conventional script notation [(S)phere, (C)ylinder, axis ($\alpha$)] were therefore converted to power vector coordinates using the following formulas:

$$J_o = \frac{-C}{2}\cos(2\alpha)$$

$$J_{45} = \frac{-C}{2}\sin(2\alpha)$$

The subjects were categorized based on the mean spherical equivalent (MSE) power of their less ametropic eye, in increments of 1.0 D. The analysis utilized a non-vectorial method to minimize interference between the magnitudes of spherical and cylindrical powers in the results, as can occur when using vectorial methods such as the power vector or power matrix methods [18].

## Statistical analysis

The data was statistically analyzed using Microsoft Excel (the Hamburg Refractive Database) and the SPSS software suite [20]. Correlations between anisometropia and ametropia were always based on the less ametropic eye. Descriptive statistics were calculated including the median, quartiles, mean, and standard deviation. Given the non-normal distribution of the data, nonparametric tests were used throughout the study. The Kruskal–Wallis test (for continuous independent variables) and the Mann–Whitney test (for binary independent variables) were used for comparisons between groups. A value of $p < 0.05$ was considered to be statistically significant.

We conducted separate analyses for subjective and cycloplegic refractive data, including only subgroups with a sample size of over 30 subjects. Logistic regression was used to identify risk factors for anisometropia, with the presence or absence of anisometropia ($\geq$1.00 D) as the dependent variable; age, spherical ametropia, cylindrical ametropia (continuous); and sex (binary) as the independent variables. The analysis was performed separately for myopic and hyperopic subjects and for subjects of all ages alongside a subset of those aged 20–40 years. To identify risk factors that were independently associated with anisometropia in this population of refractive surgery candidates, binomial logistic regression analysis was performed, thus quantitatively yielding in how far the odds to suffer from anisometropia change depending on the explanatory variables (odds ratio; OR). The logistic regression model was also used to compute the change in value of an explanatory variable needed to double/halve the odds for anisometropia ("k"-value).

These models were used to examine the associations in the whole sample and in subjects before the onset of cataract-induced refractive changes, as proposed by Qin et al. [14]. To evaluate these associations more closely, additional logistic regression models were computed in which age was analyzed as a categorical variable with 10-year intervals. In these groups, ORs for anisometropia were calculated for each decade compared with the preceding decade in subjects ranging in age from >20 to >60 years—with adjustment for spherical ametropia, cylindrical ametropia, and sex—whereas other variables were controlled for.

## Results

### Demographics and prevalence of anisometropia

The data of the refractive state of 134,603 individuals (269,206 eyes) was included in the analysis. The average age of the participants was 38.0 years (range 18–75). Of the total data set, 108,705 subjects (~81%) were myopic, and 25,898 subjects (~19%) were hyperopic.

For the largest part of the study population (95,831 individuals; ~71%), only data for subjective refraction was available. Therefore, the statistical analysis of this subgroup was limited to subjective refraction. We also measured both cycloplegic and subjective refraction for a subset of 38,772 myopic patients (~36% of the myopic subgroup) and 3,307 hyperopic patients (~13% of the hyperopic subgroup). We included an additional statistical analysis on cycloplegic refraction in this subgroup.

The median level of anisometropia determined subjectively ($A_{subj}$) in the entire study population was 0.38 D, with a prevalence of 17.9% ($A_{subj} > 1$ D). The 95% confidence interval was 0.12–0.75 D (subsequently, the 95% confidence interval will be provided in parentheses). Mild anisometropia ($1.00 \leq D < 2.00$ D) was found in 13.2%, moderate anisometropia ($2.00 \leq D < 3.00$ D) in 3.0%, and severe anisometropia ($\geq 3.00$ D) in 1.6% of the individuals. For both the myopic and hyperopic subgroups, the median level of $A_{subj}$ was found to be identical across the whole study population (0.38 D; 0.12–0.75 D). The prevalence of $A_{subj}$ for myopes was 17.7%, and the prevalence of $A_{subj}$ for hyperopes was 18.6%. Table 1 shows the clinical characteristics and descriptive statistics of the parameters for myopic and hyperopic participants, which are always based on the less ametropic eye.

**Table 1. Demographics of the study population (without/with anisometropia; entire study population).**

| | $A_{subj}$ | | | | Total | |
| --- | --- | --- | --- | --- | --- | --- |
| | <1 D | | ≥1 D | | | |
| **Patient data** | **Number** | **%** | **Number** | **%** | **Number** | **%** |
| Patients | (N = 110,562) | | (N = 24,041) | | (N = 134,603) | |
| **Sex** | | | | | | |
| Female | 61,587 | 55.7% | 14,231 | 59.2% | 75,818 | 56.3% |
| Male | 48,975 | 44.3% | 9,810 | 40.8% | 58,785 | 43.7% |
| **Refractive group** | | | | | | |
| Myopia | 89,474 | 80.9% | 19,231 | 80.0% | 108,705 | 80.8% |
| Hyperopia | 21,088 | 19.1% | 4,810 | 20.0% | 25,898 | 19.2% |
| | Range | Mean (SD) | Range | Mean (SD) | Range | Mean (SD) |
| **Myopia** | | | | | | |
| Eyes | (N = 178,948) | | (N = 38,462) | | (N = 217,410) | |
| **Age (years)** | [18.00, 74.95] | 35.01 (9.77) | [18.00,74.95] | 36.88 (10.68) | [18.00, 74.95] | 35.34 (9.96) |
| **Spherical equivalent (D)** | [−11.88, −0.11] | −3.53 (2.09) | [−13.50, −0.12] | −4.64 (2.32) | [−13.50, −0.11] | −3.72 (2.18) |
| **Sphere (D)** | [−10.00, 4.75] | −3.07 (2.13) | [−10.00, 3.00] | −4.04 (2.34) | [−10.00, 4.75] | −3.24 (2.20) |
| **Cylinder (D)** | [−9.75, 0.00] | −0.91 (0.89) | [−9.75, 0.00] | −1.19 (1.10) | [−9.75, 0.00] | −0.96 (0.94) |
| **Hyperopia** | | | | | | |
| Eyes | (N = 42,176) | | (N = 9,620) | | (N = 51,796) | |
| **Age (years)** | [18.00, 75.00] | 48,27 (12.00) | [18.00, 74.91] | 44.22 (12.70) | [18.00, 75.00] | 47.52 (12.23) |
| **Spherical equivalent (D)** | [0.00, 8.25] | 2.08 (1.48) | [0.00, 8.25] | 2.79 (1.71) | [0.00, 8.25] | 2.21 (1.55) |
| **Sphere (D)** | [0.00, 8.25] | 2.57 (1.65) | [0.00, 8.25] | 3.38 (1.86) | [0.00, 8.25] | 2.72 (1.72) |
| **Cylinder (D)** | [−9.50, 0.00] | −0.98 (1.11) | [−8.75, 0.00] | −1.19 (1.22) | [−9.50, 0.00] | −1.02 (1.13) |

The standard deviation is abbreviated as "SD."

## Subjective refraction versus cycloplegic refraction

Within the subset of patients with measurements for both $A_{subj}$ and $A_{cycl}$, the median level of anisometropia determined through cycloplegic autorefraction ($A_{cycl}$) was 0.38 D (0.12–0.75), and the overall prevalence of $A_{cycl} > 1$ D was 15.7%. For $A_{subj}$, the median level was 0.38 (0.12, 0.62) and overall prevalence 15.7%.

The prevalence of $A_{cycl}$ in myopic subjects was 15.3% and in hyperopic subjects 19.8%. This was very similar to the prevalence of $A_{subj}$ in myopic (15.2%) and hyperopic (19.8%) subjects.

## Associations between anisometropia and spherical ametropia

Fig 1(A) to 1(H) demonstrates the extent and severity of anisometropia in both myopic and hyperopic individuals. The myopic and hyperopic subjects were divided into separate groups based on the spherical power of their less ametropic eye, with intervals of 1.0 D, as first described by Qin et al. [14].

To determine if there was a significant relationship between anisometropia and ametropia, we first graphically analyzed the prevalence and level of anisometropia in relation to the manifest and cycloplegic SE power in the subject's less ametropic eye. We observed an increase in anisometropia prevalence with increasing myopia in the range of $-1.00 \geq$ D $> -6.00$ and a plateau in the range of $-6.00 \geq$ D $> -9.00$. We also observed elevated anisometropia in patients with very low myopia (D $> -1.00$) and a decrease of anisometropia prevalence in patients with very high myopia (D $< -9.00$). The picture changes when focusing solely on patients with severe anisometropia ($\geq 3.00$ D), where a decrease of anisometropia with increasing myopia can be observed, especially with elevated levels of myopia $\geq 6.00$ D (Fig 1A [$A_{subj}$] and Fig 1E [$A_{cycl}$]).

In hyperopes ($> +1.00$ D), we observed a relatively constant prevalence of $A_{subj}$ (Fig 1C) and a decreasing prevalence of $A_{cycl}$ (Fig 1G) with increasing ametropia, while we observed an overall decline of anisometropia severity with increasing hyperopia (Fig 1C and 1G).

To gain a statistical understanding of the association between anisometropia ($A_{subj}$) and spherical ametropia, we analyzed the median and average rank of anisometropia in each category of refractive error. An association was evident in myopes (Fig 1B; Kruskal–Wallis chi-square test: $\chi2 = 1,521.6$, degrees of freedom [$df$] = 9, p $< 0.001$). An association was also found in hyperopes (Fig 1D; Kruskal–Wallis test: $\chi2 = 104.71$, $df = 4$, p $< 0.001$).

When applying binomial logistic regression, the general trend of increasing anisometropia with increasing myopia could be confirmed (OR 0.918/D, 95% CI 0.912–0.925). Note that an increase in myopia results in more negative values for D—i.e., the positive relation of myopia and anisometropia is shown by an OR value $<1$. An increase in myopia by 8.15 D doubles the risk of suffering from anisometropia (k = 8.15).

In hyperopic subjects, however, logistic regression yielded an inverse relation of spherical power with anisometropia (OR 0.959/D, 95% CI 0.938–0.981). A spherical refractive error increase by 16.72 D led to an approximately twofold decrease in the OR for anisometropia (k = 16.72).

## Associations between anisometropia and cylindrical power

To further examine astigmatic refractive error, both vectorial (MSE, J0, and J45) and non-vectorial (spherical and cylindrical powers) methods were employed. In the vectorial analysis, the prevalence and severity of anisometropia changed with the level of astigmatism, with a similarly shaped distribution for the J0 and J45 cylinders. For J0, we found a V-shaped prevalence and severity of anisometropia with a minimum of J0 shifted toward the group $\leq 0$ (Fig 2A and 2E). There was an association between J0 and the observed level of anisometropia in the less

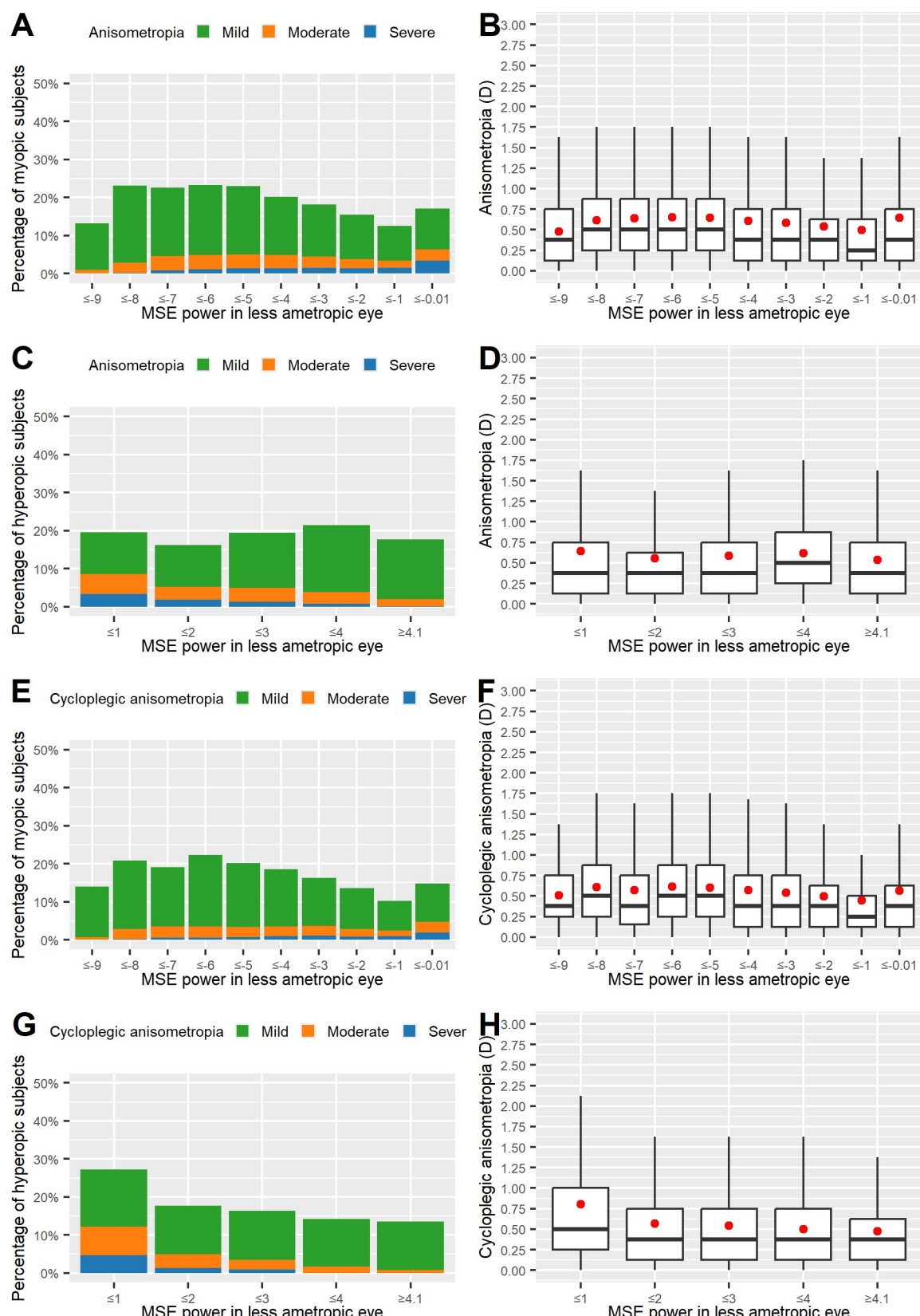

**Fig 1.** Prevalence and severity of $A_{subj}$ (A, B, C, D) and $A_{cycl}$ (E, F, G, H) and the extent of spherical ametropia (mean spherical equivalent [MSE] power). Prevalence of anisometropia for myopes (A, E) and hyperopes (C, G) and severity of anisometropia for myopes (B, F) and hyperopes (D, H). Subjects are grouped in 1.00 D intervals of MSE power in the less ametropic eye.

ametropic eye (Kruskal–Wallis test: $\chi 2 = 984.42$, $df = 4$, p < 0.001). For J45, we also found a V-shaped prevalence of anisometropia with a minimum J45 shifted toward the group ≤0, yet the entire distribution appeared more symmetrical around 0 D, with the prevalence in the group ≤ 1 being only moderately higher than the minimum in the group ≤0 (Fig 2B and 2F). As observed for J0, the Kruskal–Wallis test also showed an association between J45 and the observed level of anisometropia in the less ametropic eye ($\chi 2 = 335.43$, $df = 4$, p < 0.001).

An association between anisometropia and cylindrical power was also evident in the non-vectorial analyses. In the eyes of myopes, the prevalence of anisometropia increased as cylindrical power increased. The same was true of the severity of anisometropia (Fig 2C and 2G; Kruskal–Wallis test $\chi 2 = 1,466.3$, $df = 4$, p < 0.001).

In contrast, in the eyes of hyperopes, graphically no clear trend with regard to the overall prevalence of anisometropia was seen as cylindrical power increased, whereas the severity of anisometropia showed a decrease with increasing levels of hyperopia (Fig 2D and 2H). The Kruskal–Wallis test showed an association between anisometropia and cylindrical power for hyperopes as well ($\chi 2 = 97.74$, $df = 4$, p < 0.001).

In myopic subjects, logistic regression showed that cylindrical power was a strong risk factor for anisometropia; a cylindrical power change by −2.15 D led to an approximately twofold increase in the OR for anisometropia (OR 0.725/D; 95% CI 0.713–0.736).

In hyperopic subjects, cylindrical power (OR 1.093/D, 95% CI 1.057–1.130) was found to be inversely related to anisometropia, with a cylindrical refractive error increase by 7.8 D, reducing the risk to suffer from anisometropia by half.

## Associations between anisometropia and age

The combined group of hyperopic and myopic individuals demonstrated roughly steady prevalence (Fig 3A) and severity (Fig 3D) of anisometropia with increasing age. However, an association was shown by the Kruskal–Wallis test ($\chi 2 = 225.02$, $df = 5$, p < 0.001).

In contrast, myopic subjects displayed a continuous rise in the prevalence of anisometropia with age (Fig 3B). Severity also increased (Fig 3E; Kruskal–Wallis test: $\chi 2 = 681.79$, $df = 5$, p < 0.001).

On the other hand, in hyperopes, the prevalence of anisometropia decreased with increasing age (Fig 3C). A decrease can also be seen in the severity (Fig 3F; Kruskal–Wallis test: $\chi 2 = 694.19$, $df = 5$, p < 0.001).

As a result of the logistic regression analysis, in myopic subjects, advancing age (OR 1.018/year; 95% CI 1.017–1.020) was positively related to anisometropia, whereas in hyperopic subjects, age (OR 0.970/year, 95% CI 0.970–0.976) was found to be inversely related to anisometropia.

## Associations between anisometropia and sex

The overall prevalence of anisometropia was higher among female subjects (18.8%) than among male subjects (16.7%), and this difference was statistically significant (Mann–Whitney test W = 2317176359; p < 0.001). Although the median level of 0.38 D and distribution parameters of quartiles (Fig 4C and 4D; Q25–Q75 0.12–0.75) of anisometropia were equal between male and female subjects, the difference in mean ranks of the sex groups was statistically significant because of higher anisometropia prevalence in female subjects.

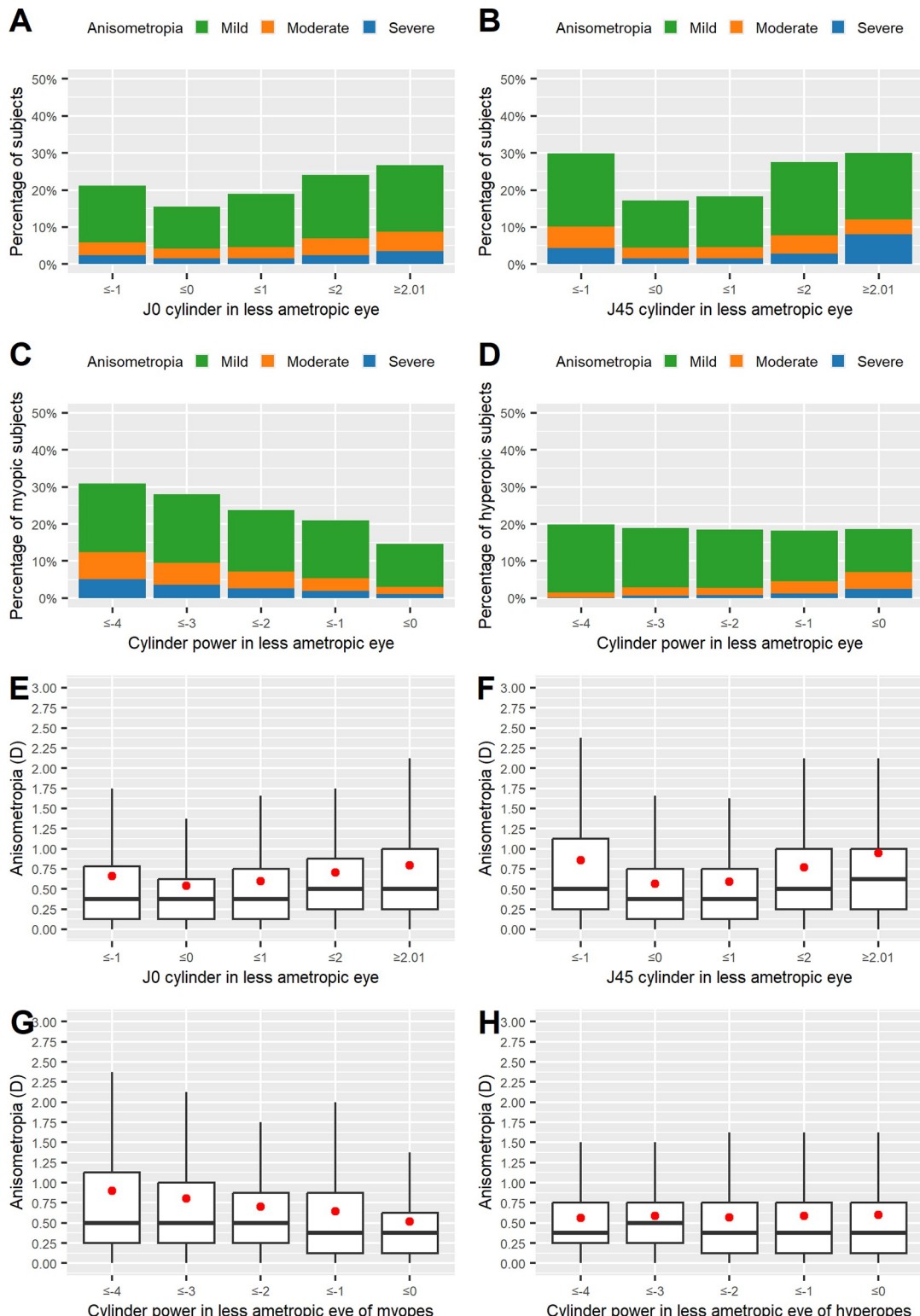

**Fig 2. Variations in the prevalence and severity of anisometropia ($A_{subj}$) in relation to cylindrical power.** For Figures A, B, E, and F, vectorial notation was used, and for Figures C, D, G, and H, spherocylindrical notation was used. The prevalence of anisometropia, differentiated in mild/medium/severe classes, dependent on the level of J0 (A), J45 (B), and their cylindrical power for myopes (C) as well as for hyperopes (D). Distribution of anisometropia severity in respective groups (E–H). Subjects are grouped in 1.00 D intervals of $J_0$, $J_{45}$, and cylindrical power in the less ametropic eye, respectively.

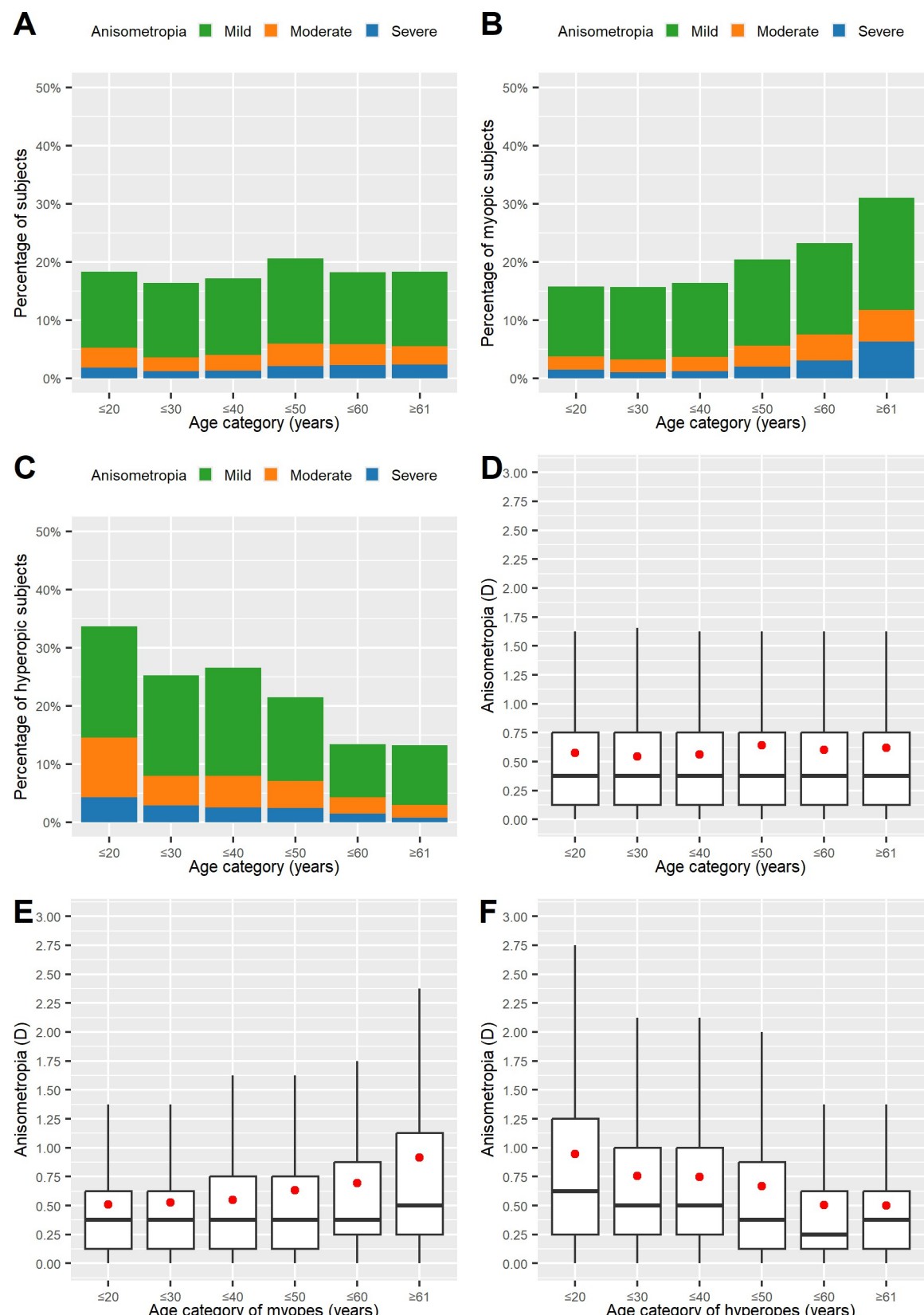

**Fig 3. Variations in the prevalence and severity of anisometropia (A$_{subj}$) in relation to age.** Prevalence of anisometropia in the whole study population (A) as well as in the myopic (B) and hyperopic (C) subgroups. Severity of anisometropia in the whole study population (D) and in the myopic (E) and hyperopic (F) subgroups.

The subgroup analysis showed that in myopic subjects, anisometropia prevalence was higher in female (19.0%) than in male (15.9%) subjects (Fig 4A; Mann–Whitney test W = 1513668780; p < 0.001). In hyperopic subjects, in contrast, the prevalence of anisometropia was significantly higher in male (19.4%) than in female (17.6%) subjects (Fig 4B; Mann–Whitney test W = 82341517; p < 0.001).

The logistic regression analysis yielded that in myopic subjects, male sex (OR 0.810, 95% CI 0.784–0.837) was found to be negatively related with anisometropia. In hyperopic subjects, by

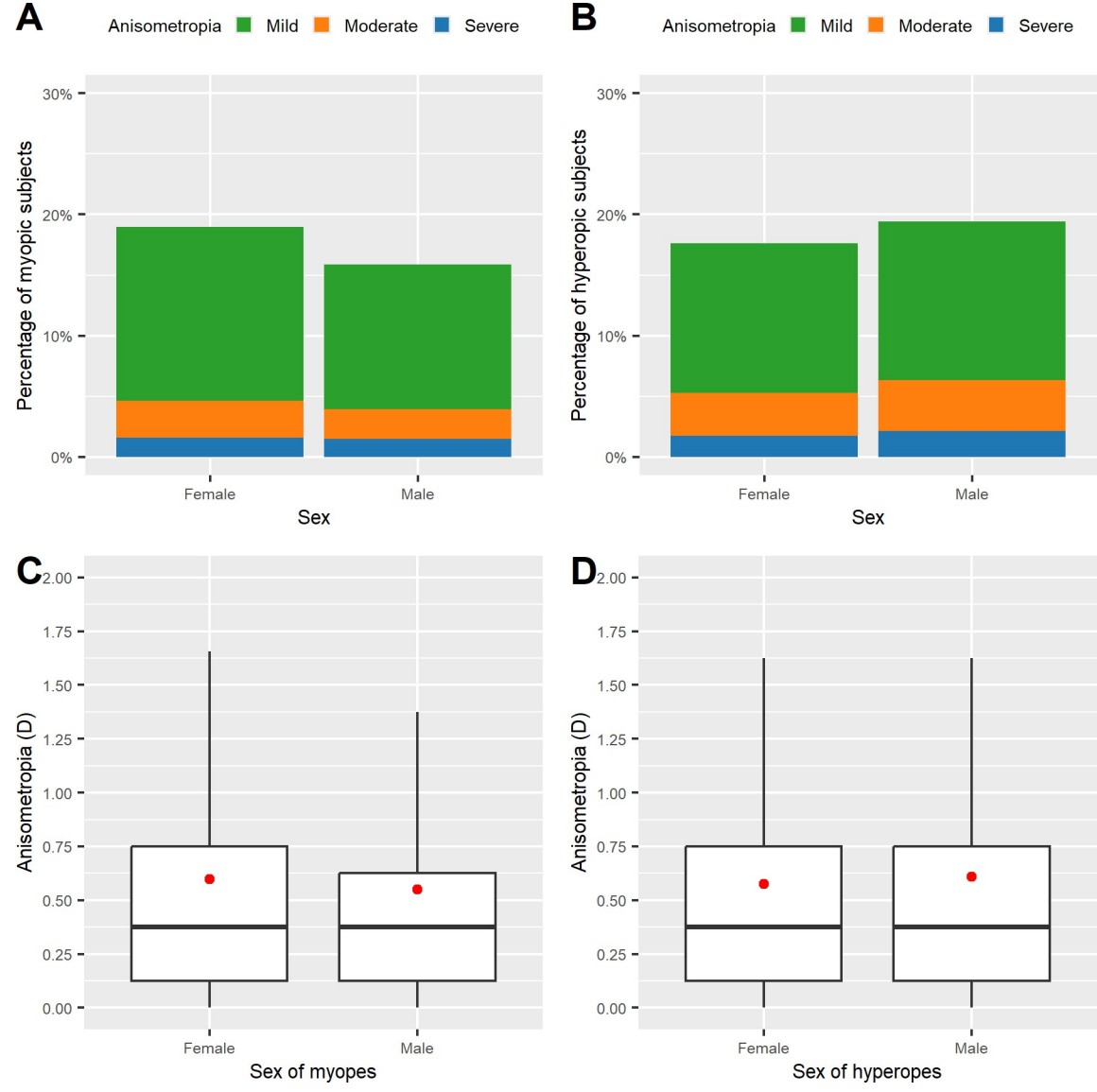

**Fig 4. Variations in the prevalence and severity of anisometropia (A$_{subj}$) in relation to sex.** Prevalence of anisometropia in myopic subjects (A) and in hyperopic subjects (B). The anisometropia severity level is shown on a grayscale. Level of anisometropia in myopes (C) and hyperopes (D).

contrast, male sex (OR 1.113, 95% CI 1.045–1.186) was found to be positively related to anisometropia.

## Summarized evaluation of the logistic regression models

In hyperopic subjects, all the explanatory variables (spherical power, cylindrical power, age, and sex) were found to be independently associated with anisometropia. Where male sex (OR 1.113, 95% CI 1.045–1.186) was found to be positively related to anisometropia, age (OR 0.970/year, 95% CI 0.970–0.976), spherical power (OR 0.959/D, 95% CI 0.938–0.981), and cylindrical power (OR 1.093/D, 95% CI 1.057–1.130; note: all data in minus cylinder form!) were found to be inversely related to anisometropia. A spherical refractive error increase by 16.72 D led to an approximately twofold decrease in the OR for anisometropia, and a cylindrical refractive error decrease by 7.8 D also led to an approximately twofold decrease in the OR for anisometropia.

In the hyperopic age group 20–40 years, in comparison, the effects of changes in the spherical and cylindrical power on the OR for anisometropia were more pronounced, with a spherical refractive error increase by 2.72 D leading to an approximately twofold decrease in the OR for anisometropia. A cylindrical refractive error increase by 3.37 D (minus cylinder) led to an approximately twofold decrease in the OR for anisometropia. In contrast to the overall data set for hyperopes, age was not independently associated with anisometropia in the age group 20–40 years.

In myopic subjects, as is the case for hyperopic subjects, all the explanatory variables (spherical power, cylindrical power, age, and sex) were independently associated with anisometropia. Cylindrical power was most strongly related with anisometropia; a cylindrical power change by −2.15 D led to an approximately twofold increase in the OR for anisometropia. Advancing age (OR 1.018/year; 95% CI 1.017–1.020, for the full age range of subjects) as well as increasing spherical (OR 0.918/D; 95% CI 0.912–0.925) and cylindrical power (OR 0.725/D; 95% CI 0.713–0.736) correlated positively (note: negative values for D!) with anisometropia in myopic subjects. As opposed to hyperopes, female sex was found to be positively related with anisometropia. A separate analysis of the age group 20–40 years did not yield fundamentally different results from those of the analysis of the entire data set for myopes.

Table 2 lists the parameters of the logistic regression models describing the associations between anisometropia and the explanatory variables spherical power, cylindrical power, age, and sex.

The association between anisometropia and age appeared to be complex and to differ between myopic and hyperopic subjects. In myopic refractive surgery candidates, a steady increase in the OR with each decade of advancing age from the fourth decade onward was observed (Fig 3B). By contrast, a steady decrease in the OR could be observed starting from the fourth decade in hyperopic refractive surgery candidates (see Fig 3C). The results from the binomial logistic regression analysis for the presence of anisometropia with age modeled as a categorical variable are presented in Table 3.

## Discussion

### Selection bias and limitations

We based this study on concepts laid out in the research carried out by Linke et al. [16], which was performed over 10 years ago, and used a significantly expanded data base for increased robustness of the results. In terms of robustness, most importantly, the extent to which the statistically significant results obtained here can be generalized with regard to anisometropia in the population as a whole should be discussed. To this end, how far the study population could

**Table 2. Binomial logistic regression analysis for the presence of anisometropia (1.0 D difference in MSE of both eyes) with age modeled as a continuous variable.**

| Model | Variable | Regression coefficient | SE of coefficient | Significance | OR | Lower | Upper | k |
|---|---|---|---|---|---|---|---|---|
| Hyperopes | Spherical power | −0.041 | 0.011 | <0.001 | 0.959 | 0.938 | 0.981 | 16.72 |
| All ages (n = 25,898) | Cylindrical power | 0.089 | 0.017 | <0.001 | 1.093 | 1.057 | 1.130 | 7.8 |
| Nagelkerke R-squared = 0.03 | Age (years) | −0.030 | 0.001 | <0.001 | 0.970 | 0.967 | 0.973 | 22.76 |
| | Sex, male | 0.107 | 0.032 | 0.001 | 1.113 | 1.045 | 1.186 | ‡ |
| | Constant | 0.062 | 0.079 | NSD | 1.064 | 0.911 | 1.242 | - |
| Myopes | Spherical power | −0.085 | 0.004 | <0.001 | 0.918 | 0.912 | 0.925 | 8.15 |
| All ages (n = 108,705) | Cylindrical power | −0.322 | 0.008 | <0.001 | 0.725 | 0.713 | 0.736 | 2.15 |
| Nagelkerke R-squared = 0.04 | Age (years) | 0.018 | 0.001 | <0.001 | 1.018 | 1.017 | 1.020 | 38.28 |
| | Sex, male | −0.210 | 0.017 | <0.001 | 0.810 | 0.784 | 0.837 | ‡ |
| | Constant | −2.688 | 0.035 | <0.001 | 0.068 | 0.064 | 0.073 | - |
| Hyperopes | Spherical power | −0.255 | 0.020 | <0.001 | 0.775 | 0.746 | 0.805 | 2.72 |
| Age 20–40 years (n = 6,138) | Cylindrical power | 0.206 | 0.026 | <0.001 | 1.228 | 1.167 | 1.294 | 3.37 |
| Nagelkerke R-squared = 0.09 | Age (years) | −0.003 | 0.005 | NSD | 0.997 | 0.986 | 1.008 | ‡ |
| | Sex, male | 0.131 | 0.061 | 0.031 | 1.140 | 1.012 | 1.284 | ‡ |
| | Constant | 0.010 | 0.184 | NSD | 1.010 | 0.704 | 1.449 | - |
| Myopes | Spherical power | −0.092 | 0.005 | <0.001 | 0.912 | 0.904 | 0.921 | 7.52 |
| Age 20–40 years (n = 74,814) | Cylindrical power | −0.337 | 0.010 | <0.001 | 0.714 | 0.700 | 0.728 | 2.06 |
| Nagelkerke R-squared = 0.03 | Age (years) | 0.006 | 0.002 | 0.002 | 1.006 | 1.002 | 1.010 | ‡ |
| | Sex, male | −0.256 | 0.021 | <0.001 | 0.774 | 0.742 | 0.807 | ‡ |
| | Constant | −2.362 | 0.066 | <0.001 | 0.094 | 0.083 | 0.107 | - |

‡ Variables not exerting sufficient influence to produce a twofold increase in the odds ratio (OR).

Independent variables for which no significant difference has been observed are abbreviated as "NSD."

**Table 3. Binomial logistic regression analysis for the presence of anisometropia (1.0 D difference in MSE of both eyes) with age modeled as a categorical variable.**

| Model | Variable | Regression coefficient | SE of coefficient | Significance | OR | Lower | Upper |
|---|---|---|---|---|---|---|---|
| Hyperopes | Spherical power | −0.048 | 0.011 | <0.001 | 0.954 | 0.933 | 0.975 |
| All ages (n = 25,898) | Cylindrical power | 0.090 | 0.017 | <0.001 | 1.095 | 1.059 | 1.132 |
| Nagelkerke R-squared = 0.04 | Sex, male | 0.098 | 0.033 | 0.003 | 1.103 | 1.035 | 1.176 |
| | Age 20–30 (years) | −0.393 | 0.133 | 0.003 | 0.675 | 0.522 | 0.880 |
| | Age 30–40 (years) | −0.359 | 0.133 | 0.007 | 0.699 | 0.540 | 0.909 |
| | Age 40–50 (years) | −0.710 | 0.131 | <0.001 | 0.491 | 0.382 | 0.637 |
| | Age 50–60 (years) | −1.324 | 0.131 | <0.001 | 0.266 | 0.206 | 0.345 |
| | Age 60–75 (years) | −1.339 | 0.137 | <0.001 | 0.262 | 0.201 | 0.344 |
| | Constant | −0.441 | 0.133 | 0.001 | 0.644 | 0.495 | 0.832 |
| Myopes | Spherical power | −0.085 | 0.004 | <0.001 | 0.919 | 0.912 | 0.926 |
| All ages (n = 108,705) | Cylindrical power | −0.323 | 0.008 | <0.001 | 0.724 | 0.713 | 0.736 |
| Nagelkerke R-squared = 0.04 | Sex, male | −0.213 | 0.017 | <0.001 | 0.808 | 0.782 | 0.835 |
| | Age 20–30 (years) | −0.047 | 0.069 | NSD | 0.954 | 0.836 | 1.093 |
| | Age 30–40 (years) | 0.000 | 0.069 | NSD | 1.000 | 0.876 | 1.145 |
| | Age 40–50 (years) | 0.256 | 0.069 | <0.001 | 1.291 | 1.129 | 1.482 |
| | Age 50–60 (years) | 0.447 | 0.071 | <0.001 | 1.563 | 1.361 | 1.801 |
| | Age 60–75 (years) | 0.870 | 0.090 | <0.001 | 2.386 | 2.000 | 2.851 |
| | Constant | −2.130 | 0.069 | <0.001 | 0.119 | 0.104 | 0.136 |

be distorted by the fact that these were candidates for refractive surgery must be logically questioned.

First of all, it should be mentioned that the study population consists only of people with ametropia who have therefore opted for refractive surgery and is therefore not representative of the population as a whole. Our study demographics are thus not representative of the overall population, and both prevalence and severity of anisometropia can be expected to be higher than that of the overall population. However, the extent to which the results are at least representative of the overall population with ametropia should be discussed. Furthermore, the clinics were not selected through defined epidemiological sampling, leading to a lack of representation from both young individuals under 18 years old and those over 75 years old. This study is thus limited to an age range of 18–75 years and does not allow any conclusions for children, adolescents, or elderly people.

With regard to the impact on the statistical analysis as such, the question on the extent to which the mere fact that patients opt for refractive surgery and not for a visual aid is already influenced by anisometropia must be asked. From a clinical perspective, we consider this to be unlikely, at least in the case of moderate refractive errors. However, particularly in the case of severe refractive error and the simultaneous occurrence of anisometropia, we consider it possible that patients are more likely to decide against refractive surgery. We discuss this in detail in the discussion section "Associations between anisometropia and spherical ametropia" (further below).

A separate question to be considered is the extent to which the presence of lower or higher spherical or cylindrical ametropia, sex, or age would have an influence on the decision for or against refractive surgery, irrespective of the degree of anisometropia. The influences are evident here, but they do not have any negative effects on our statistical evaluations. For example, women (56.3%) are overrepresented in the study as they opt for refractive surgery more frequently than men, yet it would only be problematic for the significance of our study if the presence of anisometropia played a different role in the decision for or against surgery in women than in men. We consider this unlikely. However, a distortion would occur if parameters that were not investigated had an influence on the degree of anisometropia, the parameters themselves being associated with the decision for or against refractive surgery. This would, for instance, be the case if the income in the study population, which is probably higher than the average income, itself had an influence on the degree of anisometropia. The latter was not investigated in the study and therefore cannot be ruled out. Likewise, it cannot be ruled out that the presence of anisometropia itself has an influence on the decision to undergo refractive surgery. It is conceivable, for example, that aesthetic aspects play a role. This is a limitation of the present study, in particular with regard to the prevalence of anisometropia, which may be different in the study population compared to the entire ametropic population for the aforementioned reasons. There are undoubtedly also restrictions with regard to geography: generalizations are only possible for the ametropic population in Germany, taking into account the limitations described above.

Nevertheless, we are convinced that our investigations allow us to draw conclusions about the relationships between anisometropia and the independent variables spherical refractive error, cylindrical refractive error, age, and sex in so far as they are not affected by these limitations. In the following section, our results are discussed in detail with this in mind.

## Demographics and prevalence of anisometropia

Compared to previous studies carried out in the area of anisometropia research, we included a very large amount of patient data (in total: 134,603 refractive surgery candidates) and are not

aware of any studies with a more extensive data base. This generally leads to statistically more significant results irrespective of the potential clinical significance. Against this background, we found it interesting to compare the prevalence of anisometropia with previous studies with a similar data base (refractive surgery candidates in Germany, cf. Linke et al. [16]) on the one hand and a diverging data base (optometry patients in the United Kingdom, cf. Qin et al. [14]) on the other hand. Results regarding the overall prevalence of anisometropia in the observed patient data set in principle confirm the trends found by Linke et al. over 10 years ago [16] but showed reduced prevalence in myopes (17.9% compared to 18.6%) and increased prevalence in hyperopes (18.6% compared to 16.6%). We do not know the clinical reasons for the change in the prevalence of anisometropia in candidates for refractive surgery, but shifts compared to the previous study could also result from the limitations of the previous study, in particular the much smaller data set used there. Looking beyond Germany, the overall prevalence found by Qin et al. [14] (~17%) in 2005, who investigated anisometropia in optometry patients in the United Kingdom, is also roughly in line with our results.

More recently, Hashemi et al. [10], who studied a population of Iranian schoolchildren, found an anisometropia prevalence of 8.8% in myopes and a prevalence of 5.7% in hyperopes, whereas Zhou et al. [9], who investigated Chinese schoolchildren, found a considerably higher anisometropia prevalence, 30.4% in myopes and 17.7% in hyperopes. Wang et al. [17] focused on myopic Chinese adults and found an anisometropia prevalence of 29.6% there.

## Associations between anisometropia and spherical ametropia

Our results reveal an independent association between anisometropia and spherical power both for myopes and for hyperopes. The latter is noteworthy as Linke et al. [16] only found borderline significance when looking at hyperopes. Regarding anisometropia prevalence and severity, similar patterns were found in this study compared to that of Linke et al. [16]. Most notably, our results confirm the decreasing trend of anisometropia severity with increasing ametropia (both in myopes and hyperopes) as well as a decreasing prevalence of anisometropia in patients with higher myopia (>8 D) in the less ametropic eye. This is in direct contrast to previous findings from Qin et al. [14], who observed a roughly linear trend of increasing anisometropia prevalence with increasing myopia as well as a (nonlinear) trend of increasing anisometropia with increasing hyperopia. The most intuitive explanation for the diverging results stems from the diverging study cohorts investigated by Linke et al. [16] and us on the one hand and the study carried out by Qin et al. [14] on the other hand. Where Qin et al. [14] point out that results may be distorted by insufficiently represented groups of unilateral myopes and hyperopes in the study population selected from patients visiting optometry practices in the United Kingdom, our study and the work from Linke et al. [16] likewise are potentially affected by selection bias as described in the previous section. In particular, the results could be biased in patients with particularly complex refractive errors (such as a high degree of anisometropia with very high overall refractive error) as they may be more likely to refrain from refractive surgery and, at the same time, visit optometric practices more frequently. Both effects could explain the discrepancies between our work and the one carried out by Qin et al. [14].

That said, we consider it important to obtain a better understanding of underlying pathomechanisms leading to anisometropia, especially in cases of high myopia and hyperopia. As the underlying causes are multifactorial (e.g., Hyman [21] mentions demographic, genetic, ocular, and extrinsic factors), a potential follow-up study should be designed in a way that rules out any potentially remaining selection bias and would thus need to target the general population.

With regard to the distribution of anisometropia ($A_{subj}$) across groups of myopes/hyperopes, most importantly, our results reveal that high levels of anisometropia ($\geq$3.00 D) are less prevalent with both increasing levels of hyperopia and myopia, which appears counterintuitive yet has been shown to be statistically robust. It seems conceivable to us that in the presence of severe ametropia, other pathomechanisms, in particular genetics, could play a dominant role, affecting both eyes equally and potentially diminishing the effects of any factors favoring anisometropia (e.g., environmental factors). With regard to the pathomechanisms for anisometropia in severely ametropic patients compared to less severely ametropic patients, there is, in any case, a need for further clinical research in view of our results.

When comparing our results with previous findings, for myopes, the increasing overall prevalence of anisometropia ($\geq$1.00 D) with increasing myopia is in line with the findings of Qin et al. [14] and partly with those of Linke et al. [16], who reported a maximum of anisometropia prevalence at myopia levels of $-$7.00 D with a decreasing anisometropia prevalence at very high levels of myopia ($<-$8.00 D), whereas our data showed only a less pronounced decline with myopia levels $<-$9.00 D. When assessing anisometropia severity in myopes, our results differ from those of Qin et al. [14], who present a roughly linear positive trend of increasing anisometropia severity with increasing myopia. By contrast, Linke et al. [16] observed decreasing levels of severe anisometropia ($\geq$3.00 D) with rising levels of anisometropia, which is in line with our findings.

In hyperopes, Qin et al. [14] revealed a trend of increasing prevalence of anisometropia with increasing hyperopia (yet less linear than in myopes) and no clear picture on severity, whereas Linke et al. [16] revealed a linear trend of decreasing prevalence and severity of anisometropia with increasing hyperopia. This compares with our findings, which do not provide a clear trend with respect to anisometropia prevalence but which do provide a clear trend of decreasing anisometropia severity with rising levels of hyperopia, the latter in line with the findings of Linke et al. [16].

## Associations between anisometropia and cylindrical power

Our results reveal an independent association between anisometropia and $J_0$ as well as $J_{45}$ (vectorial notation).

When the non-vectorial approach was used and the data set was split into myopic and hyperopic groups, there was an independent association between anisometropia and cylindrical power for both myopes and hyperopes. There was also evidence of an independent association between anisometropia and cylindrical power for hyperopes. This result could not be shown previously with the smaller data set used by Linke et al. [16].

There was a clear link between higher prevalence of or elevated levels of anisometropia and higher cylindrical power for myopes. This finding is perfectly in line with previous findings from Qin et al. [14] and Linke et al. [16]. The picture was less clear in hyperopes (Fig 2). It is of note that the independent association between spherical power and anisometropia remained intact even in the absence of a visible increasing or decreasing trend for hyperopes as well. Interestingly, for hyperopes, our results are in contrast to those of Qin et al. [14], who revealed rising levels and severity of anisometropia with rising levels of astigmatism, but they mirror the results presented by Linke et al. [16].

## Associations between anisometropia and age

There was an independent association between anisometropia and age, both for the entire group and for hyperopes and myopes separately.

In myopes, there was an increase in anisometropia with increasing age. The intuitive assumption that this results from cataract-induced myopia must be countered by the fact that only patients in whom no cataract was observed were included in the study. We therefore assume that other pathomechanisms are responsible for the increase in anisometropia with advancing age, but we cannot rule out the possibility that undiagnosed cataracts may have influenced the results. The same holds for hyperopes, where a downward trend for both the prevalence and the severity of anisometropia with advancing age was observed.

### Associations between anisometropia and sex

An independent association between anisometropia and female sex in the entire study population was found in our data set.

In the myopic subgroup, an independent association between anisometropia and female sex was also found. Interestingly, in the hyperopic subgroup, an independent association of anisometropia with male sex was shown, in contrast with the group of myopes. The pathomechanisms that lead to these differences between myopes and hyperopes related to sex should be the subject of subsequent investigations.

## Conclusion

This extensive large-scale retrospective study reveals independent associations between anisometropia and spherical power, cylindrical power, age, and sex in myopic and hyperopic individuals scheduled for refractive surgery. Increasing age is positively related with anisometropia in myopes but negatively related in hyperopes. The analysis of sex yielded a positive relationship of female sex and anisometropia in myopes as well as a positive relationship of male sex and anisometropia in hyperopes. Further clinical research on the underlying reasons and potential correlations between spherical power, cylindrical power, age, and sex and anisometropia is indicated.

## Supporting information

**S1 File. Minimal anonymized data set necessary to replicate the study findings.**
(XLSX)

## Acknowledgments

We would like to thank CARE Vision Germany GmbH for granting us access to their patient data base. Furthermore, the authors would like to thank the participants of the 37th Congress of the DGII 2023 on March 2–4, 2023, in Weimar, Germany, where the main findings of the study were presented, for a valuable discussion that led to improvements in this publication.

## Author Contributions

**Conceptualization:** Mona Deuchert, Andreas Frings, Stephan Linke.

**Data curation:** Mona Deuchert, Vasyl Druchkiv.

**Formal analysis:** Mona Deuchert.

**Methodology:** Mona Deuchert.

**Project administration:** Andreas Frings.

**Resources:** Andreas Frings.

**Supervision:** Andreas Frings.

**Validation:** Andreas Frings.

**Visualization:** Mona Deuchert.

**Writing – original draft:** Mona Deuchert.

**Writing – review & editing:** Andreas Frings, Jakob Schweighofer, Sajjad Muhammad, Stephan Linke, Toam Katz.

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
