## [Decision Letter · Decision Letter 0]

28 Jun 2024

PONE-D-24-18058Prevalence and Associations of Anisometropia with Spherical Ametropia, Cylindrical Power, Age, and Sex, Based on 134,603 Refractive Surgery Candidates

PLOS ONE

Dear Dr. Frings,

Thank you for submitting your manuscript to PLOS ONE. The manuscript has been assessed by our two expert reviewers, and their detailed comments are appended below. 

Reviewers raised a number of concerns regarding the unclear study methods, limited discussion, and lack of updated reference. Please carefully revise the manuscript to address all concerns raised by the reviewers.

 Please submit your revised manuscript by Aug 12 2024 11:59PM. If you will need more time than this to complete your revisions, please reply to this message or contact the journal office at plosone@plos.org. Please include the following items when submitting your revised manuscript:A rebuttal letter that responds to each point raised by the academic editor and reviewer(s). You should upload this letter as a separate file labeled 'Response to Reviewers'.A marked-up copy of your manuscript that highlights changes made to the original version. You should upload this as a separate file labeled 'Revised Manuscript with Track Changes'.An unmarked version of your revised paper without tracked changes. You should upload this as a separate file labeled 'Manuscript'.

We look forward to receiving your revised manuscript.

Kind regards,

Der-Chong Tsai, MD, PhD

Academic Editor

PLOS ONE

Reviewers' comments:

Reviewer's Responses to Questions

**Comments to the Author**

1. Is the manuscript technically sound, and do the data support the conclusions?

Reviewer #1: Yes

Reviewer #2: Yes

2. Has the statistical analysis been performed appropriately and rigorously? 

Reviewer #1: Yes

Reviewer #2: Yes

3. Have the authors made all data underlying the findings in their manuscript fully available?

Reviewer #1: Yes

Reviewer #2: Yes

4. Is the manuscript presented in an intelligible fashion and written in standard English?

Reviewer #1: Yes

Reviewer #2: Yes

5. Review Comments to the Author

Reviewer #1: The authors retrospectively investigated the rates and associated factors of anisometropia using data from 134,603 refractive surgery candidates. The study is extensively analyzed, and I would like to address several points:

1. This study involved a study population who visited refractive clinics for refractive surgery. They were either myopic (80.8%) or hyperopic, which is different from the general population. Moreover, gender, income, and many other factors may also affect the willingness to undergo refractive surgeries. The authors should be cautious when interpreting the results and should inform the readers about the impact of selection bias. For example, the prevalence and severity of anisometropia may be overestimated within the SE range of -1.0D to +1.0D. Few isometropic subjects with SE between -1.0D and +1.0D would seek refractive surgery.

2. For the definition of anisometropia, does it include a difference of 1 Diopter (≥1D) or not (>1D)? In Line 147 and Line 163, it seems that 1D is not included as anisometropia. However, in Line 87 and Line 149, 1D is classified as mild anisometropia. In Table 1, does the symbol “[1,max]” indicate that an SE difference from 1 to the maximum value is considered anisometropia?

3. In Table 1, Figure 1, Figure 2, and Figure 3, please do not use parentheses “()” or brackets “[]” to indicate ranges. Instead, use symbols such as “>”, “≥”or “<” to clearly denote the ranges.

4. What are the exclusion criteria of the study? Were subjects with previous refractive surgery or cataract surgery in any eye excluded from the analysis?

5. The results for hyperopic subgroups regarding the association of anisometropia with spherical ametropia and cylindrical power differ from the results of Qin et al. (2005) and Linke et al. (2011). Please compare these results.

Reviewer #2: The following points should be considered;

1- Please clarify certificate approval number

2- Introduction is not sufficient regarding the domain of study and what will added

3- Please clarify inclusion and exclusion criteria in details

4- Please clarify setting ( study location) and study design

5- Please clarify methods of examinations of visual acuity, IOP, refraction , pupil

6- Discussion is not sufficient regarding comparison with recent citations

7- Conclusion should be condensed

8- Please added recent references

9- Please clarify consent for publications, funding, avialbility of materials and data, abbreviations, conflict of interests, standards of reporting, ORCID of authors

10 Some typographical errors should be corrected

6. PLOS authors have the option to publish the peer review history of their article (what does this mean?). If published, this will include your full peer review and any attached files.

Reviewer #1: **Yes: **Yu-Chieh Yang

Reviewer #2: No

---

## [Author Response · Author response to Decision Letter 0]

28 Aug 2024

Dear Mr Der-Chong Tsai,

Thank you for inviting us to submit a revised draft of our manuscript entitled, "Prevalence and associations of anisometropia with spherical ametropia, cylindrical power, age, and sex, based on 134,603 refractive surgery candidates" (PONE-D-24-18058) to PLOS ONE. We also appreciate the time and effort you and each of the reviewers have dedicated to providing insightful feedback on ways to strengthen our paper. Thus, it is with great pleasure that we resubmit our article for further consideration. We have incorporated changes that reflect the detailed suggestions you have graciously provided. We also hope that our edits and the responses we provide below satisfactorily address all the issues and concerns you and the reviewers have noted.

To facilitate your review of our revisions, the following is a point-by-point response to the questions and comments delivered in your letter dated June 28, 2024:

EDITORIAL COMMENTS

 Response: We have ensured that our manuscript meets PLOS ONE's style requirements, including those for file naming.

 Response: We have obtained written consent from our patients, which were fully anonymized. This is now stated in detail in lines 85 to 91.

3. We note that your Data Availability Statement is currently as follows: [All relevant data are within the manuscript and its Supporting Information files.] Please confirm at this time whether or not your submission contains all raw data required to replicate the results of your study. Authors must share the “minimal data set” for their submission. (…)

 Response: We now included the minimal data set necessary to replicate our study findings as a xlsx-File in the Supporting Information section.

Response: We have included the full ethics statement in the ‘Methods’ section of your manuscript file. We have obtained written consent. Cf. lines 85-91

REVIEWER 1:

1. This study involved a study population who visited refractive clinics for refractive surgery. They were either myopic (80.8%) or hyperopic, which is different from the general population. Moreover, gender, income, and many other factors may also affect the willingness to undergo refractive surgeries. The authors should be cautious when interpreting the results and should inform the readers about the impact of selection bias. For example, the prevalence and severity of anisometropia may be overestimated within the SE range of -1.0D to +1.0D. Few isometropic subjects with SE between -1.0D and +1.0D would seek refractive surgery.

 Response: Thank you for your valuable comment on the selection bias. We kindly refer to lines 350 to 388 with an in-depth discussion on selection bias.

2. For the definition of anisometropia, does it include a difference of 1 Diopter (≥1D) or not (>1D)? In Line 147 and Line 163, it seems that 1D is not included as anisometropia. However, in Line 87 and Line 149, 1D is classified as mild anisometropia. In Table 1, does the symbol “[1,max]” indicate that an SE difference from 1 to the maximum value is considered anisometropia?

 Response: A difference of 1 Diopter (≥1D) is consistently included as anisometropia. We have corrected the typographical errors in line 87 (now line 111) and in line 149 (now line 173). Table 1 (lines 181-182) was insofar clarified (see also Point 3 below).

3. In Table 1, Figure 1, Figure 2, and Figure 3, please do not use parentheses “()” or brackets “[]” to indicate ranges. Instead, use symbols such as “>”, “≥”or “<” to clearly denote the ranges.

 Response: We changed Table 1 and the Figures accordingly. In the course of adjusting the figure legends, we switched to Mild/Moderate/Severe anisometropia in line with the definition we give in the manuscript text.

4. What are the exclusion criteria of the study? Were subjects with previous refractive surgery or cataract surgery in any eye excluded from the analysis?

 Response: We specified the exclusion criteria (patients with a previous surgical intervention, such as cataract surgery, or patients with an eye condition) in lines 83-84.

5. The results for hyperopic subgroups regarding the association of anisometropia with spherical ametropia and cylindrical power differ from the results of Qin et al. (2005) and Linke et al. (2011). Please compare these results.

 Response: You have of course made an interesting and important point here. We gladly added this to the discussion and compared both aforementioned studies with our results (cf. lines 390 onwards, lines 414 onwards and lines 456 onwards in the Discussion chapter).

REVIEWER 2:

1. Please clarify certificate approval number

 Response: We added the number (#2021-1278) in lines 88-89.

2. Introduction is not sufficient regarding the domain of study and what will added

 Response: We amended our introduction and expanded on the domain of study in lines 70 to 75 in the Introduction.

3. Please clarify inclusion and exclusion criteria in details

 Response: We specified the inclusion criteria in lines 78-83 and furthermore exclusion criteria (patients with a previous surgical intervention, such as cataract surgery, or patients with an eye condition) in lines 83-84.

4. Please clarify setting (study location) and study design

 Response: We amended our Methods section (lines 77 onward) to include more details on study setting and design.

5. Please clarify methods of examinations of visual acuity, IOP, refraction , pupil

 Response: We amended our Methods section in lines 92-100 to include more details.

6. Discussion is not sufficient regarding comparison with recent citations

 Response: We included three more recent citations (2020, 2023, 2024) in lines 408-412 of our Discussion.

7. Conclusion should be condensed

 Response: We revised and condensed the Conclusion (cf. Lines 495-502)

8. Please added recent references

 Response: We added 5 recent references in total (cf. lines 51-54).

9. Please clarify consent for publications, funding, avialbility of materials and data, abbreviations, conflict of interests, standards of reporting, ORCID of authors

 Response: We clarified consent for publication (cf. lines 90-91) and included additional information on funding and competing interests in a separate ‘Additional Information’ section (cf. lines 558-565). We included the minimal data set required to replicate our study in an Excel file in the Supporting Information section. Abbreviations were consistently introduced at first mention. My ORCID number: 0000-0001-8977-4261

10. Some typographical errors should be corrected

 Response: We revised the manuscript and corrected remaining typographical errors.

Once again, we extend our gratitude for the valuable insights and your dedication in reviewing our manuscript. We hope that the revised version of our manuscript will meet your approval. Should you have any further questions or require additional information, please feel free to contact us.

Sincerely,

Andreas Frings, MD

Corresponding Author

University Hospital Düsseldorf

---

## [Decision Letter · Decision Letter 1]

22 Sep 2024

PONE-D-24-18058R1Prevalence and Associations of Anisometropia with Spherical Ametropia, Cylindrical Power, Age, and Sex, Based on 134,603 Refractive Surgery CandidatesPLOS ONE

Dear Dr. Frings,

Thank you for submitting your manuscript to PLOS ONE. After careful consideration, we feel that it has merit but does not fully meet PLOS ONE’s publication criteria as it currently stands. Therefore, we invite you to submit a revised version of the manuscript that addresses the points raised during the review process.

We look forward to receiving your revised manuscript.

Kind regards,

Der-Chong Tsai, MD, PhD

Academic Editor

PLOS ONE

Journal Requirements:

Reviewers' comments:

Reviewer's Responses to Questions

**Comments to the Author**

1. If the authors have adequately addressed your comments raised in a previous round of review and you feel that this manuscript is now acceptable for publication, you may indicate that here to bypass the “Comments to the Author” section, enter your conflict of interest statement in the “Confidential to Editor” section, and submit your "Accept" recommendation.

Reviewer #1: (No Response)

2. Is the manuscript technically sound, and do the data support the conclusions?

Reviewer #1: Yes

3. Has the statistical analysis been performed appropriately and rigorously? 

Reviewer #1: Yes

4. Have the authors made all data underlying the findings in their manuscript fully available?

Reviewer #1: Yes

5. Is the manuscript presented in an intelligible fashion and written in standard English?

Reviewer #1: No

6. Review Comments to the Author

Reviewer #1: Thank you for your response. The authors have addressed most of the points. However, in the "Discussion" section, I hope the authors can provide explanations for the data rather than simply comparing the results with previous studies. In particular, I suggest elaborating on the sections regarding "selection bias" and the "associations between anisometropia and spherical ametropia." What are the potential reasons for the observed differences in data compared to previous research? Are there any possible physiological explanations, or could selection bias have influenced the results? Additionally, I recommend having the manuscript reviewed by a native English speaker to enhance the language quality.

7. PLOS authors have the option to publish the peer review history of their article (what does this mean?). If published, this will include your full peer review and any attached files.

Reviewer #1: **Yes: **Yu-Chieh Yang

---

## [Author Response · Author response to Decision Letter 1]

1 Nov 2024

Dear Mr Der-Chong Tsai,

Thank you for revising our manuscript: "Prevalence and associations of anisometropia with spherical ametropia, cylindrical power, age, and sex, based on 134,603 refractive surgery candidates" (PONE-D-24-18058 / PONE-D-24-18058R1) to PLOS ONE, a second time. 

We are grateful for the time and effort you and the reviewers have dedicated once again on our paper. We have made revisions based on the detailed suggestions you kindly provided and hope that our edits, along with the responses below, satisfactorily address all the concerns and issues raised.

To facilitate your review of our revisions, the following is a point-by-point response to the questions and comments delivered in your letter dated September 22, 2024:

JOURNAL REQUIREMENTS

 Response: We corrected some publication data in line 550 and 582 next to several typographical corrections. In lines 583-584 we have added an additional reference (Discussion chapter, see reviewer comments below) so that we now consider the reference list complete and correct. We furthermore checked that no cited article has been retracted.

REVIEWER COMMENTS

1. I recommend having the manuscript reviewed by a native English speaker to enhance the language quality.

 Response: We had the manuscript proofread and corrected by a native English speaker.

2. Thank you for your response. The authors have addressed most of the points. However, in the "Discussion" section, I hope the authors can provide explanations for the data rather than simply comparing the results with previous studies. In particular, I suggest elaborating on the sections regarding "selection bias" and the "associations between anisometropia and spherical ametropia." What are the potential reasons for the observed differences in data compared to previous research? Are there any possible physiological explanations, or could selection bias have influenced the results? 

 Response: We have comprehensively revised the chapters on “selection bias” and “associations between anisometropia and spherical ametropia”. In doing so, we have added further explanations for our results and have addressed selection bias, which could result from the structure of the study population, much more extensively than before. We have also incorporated our clinical experience into the discussion. In terms of potential physiological explanations for our results, we have added further literature and ultimately come to the conclusion that, due to the multifactorial causes of anisometropia, further research in this area on the basis of the general population is indicated in order to be able to completely rule out selection bias. However, in the discussion we also address why we nevertheless consider our results to be of high relevance.

Sincerely,

Andreas Frings, MD

Corresponding Author

University Hospital Düsseldorf

---

## [Decision Letter · Decision Letter 2]

21 Nov 2024

Prevalence and associations of anisometropia with spherical ametropia, cylindrical power, age, and sex, based on 134,603 refractive surgery candidates

PONE-D-24-18058R2

Dear Dr. Frings,

We’re pleased to inform you that your manuscript has been judged scientifically suitable for publication and will be formally accepted for publication once it meets all outstanding technical requirements.

Kind regards,

Der-Chong Tsai, MD, PhD

Academic Editor

PLOS ONE

Additional Editor Comments (optional):

Reviewers' comments:

Reviewer's Responses to Questions

**Comments to the Author**

1. If the authors have adequately addressed your comments raised in a previous round of review and you feel that this manuscript is now acceptable for publication, you may indicate that here to bypass the “Comments to the Author” section, enter your conflict of interest statement in the “Confidential to Editor” section, and submit your "Accept" recommendation.

Reviewer #1: All comments have been addressed

2. Is the manuscript technically sound, and do the data support the conclusions?

Reviewer #1: (No Response)

3. Has the statistical analysis been performed appropriately and rigorously? 

Reviewer #1: (No Response)

4. Have the authors made all data underlying the findings in their manuscript fully available?

Reviewer #1: (No Response)

5. Is the manuscript presented in an intelligible fashion and written in standard English?

Reviewer #1: (No Response)

6. Review Comments to the Author

Reviewer #1: (No Response)

7. PLOS authors have the option to publish the peer review history of their article (what does this mean?). If published, this will include your full peer review and any attached files.

Reviewer #1: **Yes: **Yu-Chieh Yang
